# Small Noncoding RNA Signatures for Determining the Developmental Potential of an Embryo at the Morula Stage

**DOI:** 10.3390/ijms21249399

**Published:** 2020-12-10

**Authors:** Angelika Timofeeva, Yulia Drapkina, Ivan Fedorov, Vitaliy Chagovets, Nataliya Makarova, Maria Shamina, Elena Kalinina, Gennady Sukhikh

**Affiliations:** Kulakov National Medical Research Center of Obstetrics, Gynecology and Perinatology, Ministry of Health of Russia, Ac. Oparina 4, 117997 Moscow, Russia; yu_drapkina@oparina4.ru (Y.D.); is_fedorov@oparina4.ru (I.F.); v_chagovets@oparina4.ru (V.C.); np_makarova@oparina4.ru (N.M.); mariashamina@mail.ru (M.S.); e_kalinina@oparina4.ru (E.K.); g_sukhikh@oparina4.ru (G.S.)

**Keywords:** miRNA, piRNA, small RNA deep sequencing, qPCR, spent culture medium, morula, blastocyst, IVF, ART

## Abstract

As part of the optimization of assisted reproductive technology programs, the aim of the study was to identify key small noncoding RNA (sncRNA) molecules that participate in maternal-to-zygotic transition and determine development potential and competence to form a healthy fetus. Small RNA deep sequencing followed by quantitative real-time RT-PCR was used to profile sncRNAs in 50 samples of spent culture medium from morula with different development potentials (no potential (degradation/developmental arrest), low potential (poor-quality blastocyst), and high potential (good/excellent quality blastocyst capable of implanting and leading to live birth)) obtained from 27 subfertile couples who underwent in vitro fertilization. We have shown that the quality of embryos at the morula stage is determined by secretion/uptake rates of certain sets of piRNAs and miRNAs, namely hsa_piR_011291, hsa_piR_019122, hsa_piR_001311, hsa_piR_015026, hsa_piR_015462, hsa_piR_016735, hsa_piR_019675, hsa_piR_020381, hsa_piR_020485, hsa_piR_004880, hsa_piR_000807, hsa-let-7b-5p, and hsa-let-7i-5p. Predicted gene targets of these sncRNAs included those globally decreased at the 8-cell–morula–blastocyst stage and critical to early embryo development. We show new original data on sncRNA profiling in spent culture medium from morula with different development potential. Our findings provide a view of a more complex network that controls human embryogenesis at the pre-implantation stage. Further research is required using reporter analysis to experimentally confirm interactions between identified sncRNA/gene target pairs.

## 1. Introduction

The progress in assisted reproductive technology (ART) is mostly focused on the development of new modified schemes for ovarian stimulation to produce an optimum number of oocytes to maximize success in the safest possible way as well as on fertilization techniques to improve embryo quality [1]. These strategies lead to better resultant embryos but with the resulting pregnancy rate from in vitro fertilization (IVF)/intracytoplasmic sperm injection (ICSI) protocols still not exceeding 30–40% [2].

Almost all types of gonadotropins are widely used for ovarian stimulation. These gonadotropins have different dosages of luteinizing hormone (LH), diverse biological activity and stability of follicle-stimulating hormone (FSH), and purification level and composition of used isoforms; however, their effectiveness does not differ significantly according to the results of various meta-analyses [3]. The effects of ovarian stimulation on endometrial receptivity and quality and number of oocytes have been studied in protocols with gonadotropin-releasing hormone (GnRH) agonists (a-GnRH) and GnRH antagonists (ant-GnRH) [4]. In patients with preserved or diminished ovarian reserve (DOR), standard protocols with ant-GnRH are used for ovarian stimulation. An alternative treatment regimen with a-GnRH is recommended for patients with preserved ovarian reserve, severe endometriosis, and/or myoma. However, in cases where there is a high risk for ovarian hyperstimulation syndrome (OHSS), this protocol is not recommended because it is not possible to replace human chorionic gonadotropin (hCG) with a-GnRH for final oocyte maturation. Other methods of OHSS prevention include dopamine agonist use, and a freeze-all strategy can be also a good option [5].

To date, ovarian stimulation protocols in in vitro fertilization programs are optimized to produce oocytes of high quality, which determines the success of fertilization and affects the consequent embryo development, pregnancy, and live birth. Unfortunately, there is a mass of oocytes that are not fertilized despite normal morphology. Moreover, even if fertilization occurs, only some fertilized oocytes fully complete their preimplantation development, and even fewer are capable of implantation [6]. Early embryonic development immediately after fertilization is directed by maternal mRNAs expressed in oocytes and stored in mRNPs [7] and enables further development to the blastocyst stage [8]. Several waves of maternal mRNAs clearance between the 4- and 8-cell stage and at the blastocyst stage should occur for activation of the embryo genome. Moreover, these waves are gene-specific and finely regulated [9,10,11]. Hierarchical clustering of RNA sequencing data from embryo samples of different developmental stages revealed similarity in both morula and 8-cell embryo mRNA expression patterns that differed dramatically from that of the blastocyst stage [9]. Therefore, studying the level of gene expression for a more detailed understanding of zygotic genome activation and prognosing further development of the embryo is most promising and convenient at the morula stage.

In recent years, scientists have paid close attention to the role of small noncoding RNAs (sncRNAs) in embryo implantation and its normal development considering their previously demonstrated multifunctional effect on the transcriptional and post-transcriptional levels of gene expression regulation [12,13,14,15,16,17]. It has been shown that upon oocyte fertilization, sperm miRNA and piRNA cleared maternal mRNAs, while sperm mRNAs are actively translated by the enzymatic apparatus of the oocyte, leading to increased expression levels of transcription factors (for example, bobby sox homolog, zinc finger protein 646) and histone modifiers such as ankyrin repeat domain-containing protein 12, and activation of the maternal-to-zygotic transition (MZT) and appropriate genome program [18].

sncRNAs are secreted by the growing embryo into the spent culture medium [19] and therefore can reflect the changes that occur during MZT, further contributing to genome activity in the embryo. In order to assess the viability of the embryo, its ploidy, and implantation potential, a number of scientific works have used quantitative analysis of the most studied class of small noncoding RNAs, miRNAs [19,20,21,22,23,24,25]. However, the discrepancies existing in the results obtained by various scientific groups when analyzing the spent culture medium can be explained by the following: (1) low sample size; (2) inability to reliably measure the total concentration of RNA in spent culture medium and assess its quality by Nanodrop spectrophotometer, Qubit fluorometer, or Agilent Bioanalyzer system which also applies to other biological fluids such as, for example, human peripheral blood plasma [26], which has prompted the search for normalizing endogenous RNA suitable for accurately quantitating microRNAs; (3) use of different types of reference RNA molecules (exogenous or endogenous) to normalize data in the detection of miRNAs by qPCR [27,28,29]; (4) use of conditioned media from different manufacturers which, prior to contact with the embryo, already contain sncRNAs in different quantitative ratios [30]; (5) differences in the efficiency of miRNA recovery, which is largely influenced by the isolation method used [31,32]; (6) applying different kits of cDNA libraries synthesis for sequencing [33]; (6) dependence of sncRNA expression on the embryo development stage and even on the morphofunctional characteristics of embryos at the same stage, for example, excellent, good, fair, and poor-quality blastocysts [23] according to the Gardner grading scale.

In a recent study, Kirkegaard K. et al. found a lack of reproducibility in the detection of miRNAs in a spent culture medium and questioned the use of miRNAs as a reliable biomarker, but suggested tiRNA fragments as potential novel biomarkers, which appeared to be overexpressed in conditioned IVF media samples [30]. In our previous study [23], we sequenced sncRNAs in the spent culture medium of an excellent blastocyst and in its blastocoelic fluid and showed the prevalence of piRNA molecules over miRNAs in the analyzed sample contents. Moreover, we have demonstrated significant differences in the detected levels of the sncRNA in spent culture media from embryos on the fourth day after fertilization, which had different rates of development, and by the fifth day, reached either the 3–7-cell embryo stage, or were morula, cavernous morula, or excellent, good, fair, and poor-quality blastocysts. In the present study, we were interested in comparing the levels of miRNAs and piRNAs in the spent culture media from embryos that had the same rates of development and reached the morula stage by the fourth day but had different developmental outcomes by the fifth day (degeneration or developmental arrest, a blastocyst of poor quality, or a blastocyst of excellent quality which, when implanted in the uterine, led to the birth of a healthy child). The choice of the morula stage for a detailed study was determined by the possibility of assessing, on the one hand, the ability of the embryo to pass the 8-cell stage at which the main MZT wave occurs and, on the other hand, the potential of the embryo to develop into the blastocyst able to be implanted for initiating pregnancy. We found it interesting to compare the list of potential target genes of the miRNAs and piRNAs that we identified with the data published by other colleagues on the expression profile of these target genes, the protein products of which are involved in MZT.

## 2. Results

### 2.1. Search for Interrelations between the ART Program Outcome, Parameters of Gametogenesis, and IVF/ICSI Protocol

According to Appendix A, the quality of the obtained embryos varied between assessed subfertile couples with different ART program results. Moreover, the prevalence (>50%) of blastocysts of excellent/good quality was observed in 9 out of 27 couples, of which only three embryo transfers led to a healthy pregnancy and live birth. At the same time, obtaining all embryos of excellent/good quality from a couple did not guarantee the onset of pregnancy (absence of pregnancy in couples 8, 16, 18). With a prevalence (>50%) of embryos not suitable for transfer on day 5 after fertilization (a.f.) (degenerating morula, morula with arrested development, and poor-quality blastocysts), pregnancy occurred in only 3 out of 11 couples, and in 2 couples (9, 12), embryo transfer was canceled due to the presence of only degraded embryos on day 5 a.f. Comparison of clinical data and characteristics of the ART program revealed statistically significant differences in couples with pregnancy onset and live birth (*n* = 11) relative to couples with a negative result of the ART program (*n* = 16) in the following parameters: a higher anti-Müllerian hormone (AMH) level (*p* = 0.0061), a shorter period of ovarian stimulation (*p* = 0.0226), lower doses of gonadotropin (*p* = 0.0101), and the preferable use of r-FSH (*p* = 0.0065) instead of human menopause gonadotrophins (HMG, *p* = 0.0025) for ovarian stimulation (Table 1).

To search for interrelations between the ART program outcome, parameters of gametogenesis, and IVF/ICSI protocol, a correlation analysis was performed.

Since both quantitative and qualitative characteristics were analyzed, the correlation analysis was performed using Spearman’s nonparametric correlation test (Figure 1). It was found that in the case of rFSH use, a lower total dose and a shorter duration for ovarian stimulation were required (r = −0.71, *p* < 0.0001 and r = −0.51, *p* = 0.008, respectively) than in case of HMG (r = 0.51, *p* = 0.0081 and r = 0.45, *p* = 0.0223, respectively), while no correlations between the use of one or another type of gonadotropin with the number of oocyte–cumulus complexes (OCCs) and the outcome of the ART program were found. Female age was negatively correlated with the number of OCCs (r = −0.46, *p* = 0.0183) and the number of metaphase II (MII) oocytes (r = −0.51, *p* = 0.0073) but positively correlated with the percentage of degenerated morula (r = 0.4, *p* = 0.0439). The AMH level was negatively correlated with female age (r = −0.58, *p* = 0.0021), use of HMG (r = −0.57, *p* = 0.0025), gonadotropin dose (r = −0.5, *p* = 0.0087), and the percentage of embryos not suitable for transfer (r = −0.4, *p* = 0.044) but positively correlated with the use of rFSH (r = 0.51, *p* = 0.0073), number of OCC (r = 0.48, *p* = 0.014), and number of oocytes at the MII stage (r = 0.49, *p* = 0.0114).

The type of ovulation trigger affected the oocyte quality and ART program outcome: the use of human chorionic gonadotropin (HCG) at a dose of 10,000 IU was positively correlated with the MII/OCC ratio (r = 0.46, *p* = 0.0184) and pregnancy onset (r = 0.5, *p* = 0.0098) in contrast to the use of 0.2 mg decapeptyl. The use of decapeptyl was negatively correlated with the MII/OCC ratio (r = −0.54, *p* = 0.0042). The number of OCCs was negatively correlated with the MII/OCC ratio (r = −0.45, *p* = 0.02). As for the parameters of spermatogenesis, sperm concentration was positively correlated with the number of progressively motile sperm (r = 0.49, *p* = 0.012) and with the number of morphologically normal forms (r = 0.52, *p* = 0.0066) and, in turn, the number of progressively motile sperm was positively correlated with the number of morphologically normal forms (r = 0.54, *p* = 0.0043). The absence of statistically significant correlation between the parameters of spermatogenesis and the outcomes of ART programs might be explained by impaired spermatogenesis in all men from couples included in the study, apart from couple 19, where the man was diagnosed with normozoospermia. Moreover, according to the literature, normozoospermia is not a reflection of sperm fertility, particularly in the case of idiopathic infertility [34]. To understand the molecular biological reasons for implantation failure when high-quality blastocysts are selected by their morphometric parameters and transferred into the uterine cavity, we analyzed the profile of sncRNAs secreted by the embryo and identified the molecules responsible for maternal–zygotic transition and subsequent blastulation, thereby determining embryo implantation and development potential.

### 2.2. Characterization of the Morula Secretome by RNA-Seq

Since morula development potential varies despite the cultivation conditions being the same, it was intriguing to analyze whether or not sncRNAs secreted into the culture medium by these embryos on the 4th day after fertilization were different. The samples of morula sncRNAs in spent culture medium for NGS analysis comprised four groups (see Materials and Methods), 3 out of 12 samples from group I, 3 out of 8 samples from group II, 4 out of 20 samples from group III, 3 out of 9 samples from group IV as well 1 sample of culture medium without contact with any embryo, incubated for 4 days at 37 °C as a Reference. There were 372 piRNAs and 87 miRNAs that were identified in at least one of the analyzed samples. It is important to note that the reference culture medium also contained some types of sncRNAs from human serum comprising the CSCM. Since there were no statistically significant differences in sncRNA read counts in samples of group III from those in samples of group IV, these two groups were combined to compare with groups of samples I or II. A summary of the sncRNAs read counts after normalization performed with the DESeq2 package is presented in Table 2. When comparing samples group I with group III and IV, statistically significant differences were found in the detected levels of 24 piRNAs and 2 miRNAs, some of which also statistically significantly distinguished group II from groups III and IV (Table 2).

### 2.3. Gene Expression Validation by Quantitative Real-Time PCR

qPCR was used to validate the sequencing data for all collected samples (*n* = 51). Of the 24 piRNAs that significantly distinguished group I from groups III and IV according to deep sequencing data, 16 were selected (highlighted in bold in Table 2). The selection of piRNAs was based on the possibility to optimize the conditions for qPCR to determine the optimal annealing temperature for the sense primer (detectable fluorescence signal with the lowest Ct and a single peak of the amplification product melting curve). Sense primers for piRNA expression analysis were synthesized in Evrogen (Russia, Moscow, http://evrogen.com/), and sense primers for analysis of hsa-let-7b-5p and hsa-let-7i-5p were ordered from Qiagen (primer sequences are shown in Table 3).

We found that the development of a morula into a competent embryo with high implantation potential is characterized by pronounced changes in the secretion/uptake rates of sncRNAs (Table 4, Figure 2). Morula which later reached the blastocyst stage with good/excellent morphological parameters (group I) secreted hsa_piR_011291, hsa_piR_001311, hsa_piR_015462, hsa_piR_016735, hsa_piR_019675, hsa_piR_020381, hsa_piR_004880 into the spent medium at a level 2.8–28 times higher than the background level in the reference culture medium without contact with any embryo (Table 4, Figure 2a). By contrast, spent culture medium from morula that subsequently stopped developing or degenerated (III, IV groups) contained hsa_piR_011291 and hsa_piR_016735 at levels only 1.6–1.8 times higher than the background level in the reference culture medium without any embryo and contained hsa_piR_001311, hsa_piR_015462, hsa_piR_019675, hsa_piR_020381, and hsa_piR_004880 in amounts that did not differ from those in the reference medium (Table 4, Figure 2a). In addition, we found that groups of morula with different potential for blastulation distinguished themselves by the detection level of sncRNAs with more pronounced release of hsa_piR_019122, hsa-let-7b-5p, and hsa-let-7i-5p into the culture medium in the case of morula from group I compared to morula from groups III or IV (Table 4, Figure 2b), and increased the level of hsa-let-7i-5p and decreased the level of hsa_piR_019675 in the case of morula from group II compared to morula from groups III or IV (Table 4, Figure 2a,b).

A decrease in the detection level of hsa_piR_015026 and hsa_piR_020485 was observed in spent culture medium collected from morula of groups III and IV in contrast to group I, in which there were no changes in the detection level of these molecules (Figure 2c). A decrease in the detection level of hsa_piR_000807 was observed in the morula culture medium of all groups, with a more pronounced decrease in groups III and IV as compared to the reference medium (Figure 2c). Moreover, we found that some of the above molecules were associated not only with the morula’s potential for blastulation but with the quality of the blastulation itself. In particular, we found a statistically significant increased detection level of hsa_piR_015462, hsa_piR_019675, hsa_piR_020381, and hsa_piR_004880 in the spent culture medium from morula that later developed into a good/excellent blastocyst (group I) in comparison with morula that developed into a poor blastocyst (group II) (Table 4).

Thus, morula with the ability to develop to a blastocyst of good/excellent quality with high implantation potential are characterized not only by the release but also the uptake of certain species of sncRNAs. Abnormality of processes involving these sncRNAs, by causes not established in this work, leads to the formation of a blastocyst of poor quality or even halted development or degradation. The increased release of the sncRNAs mentioned above by morula is a reflection of the activation of the zygotic genome occurring at the 4–8 cell stage and is necessary for embryogenesis and the establishment of pregnancy.

### 2.4. Identification of piRNA and miRNA Targets

The identified piRNAs were mapped to transposons, tRNA and rRNA species, and individual mRNA transcripts according to PiRBase data (http://www.regulatoryrna.org/database/piRNA/search.php), which is a reflection of their function, in particular, participation in transposon silencing to safeguard genome integrity [35] and regulation of the translation machinery through tRNA-derived small RNA accumulation, which is associated with high proliferative index of cells and the prevention of apoptosis [36]. Due to the lack of a database on target genes of human piRNAs, in order to predict the possible targets of piRNAs, we used the GRCh38 database to download RefSeq transcript sequences (https://www.ncbi.nlm.nih.gov/genome/guide/human/) and the miRanda algorithm with the alignment score of sc ≥ 170 and binding energy of en ≤ −20.0 kcal mol^−1^, as described by Roy J et al. [37], followed by selection of targets based on the degree of complementarity to seed sites of piRNAs, namely perfect matching to nucleotides 2–11 (primary seed) and a maximum of four mismatches being tolerated in nucleotides 12–21 (secondary seed), as described by Goh WS et al. [38], using scripts written in the R language (Appendix A). We converted RefSeq mRNA accessions to gene symbols using the bioDBnet database (https://biodbnet-abcc.ncifcrf.gov/db/db2db.php) for mRNAs that were potential targets for overexpressed piRNAs in the morula group (group I) with high potential for blastulation and implantation, that is, for all piRNAs indicated in Table 4 except hsa_piR_020485 and hsa_piR_000807. The list of RNA targets for these piRNAs is presented in Appendix A (Sheet 1).

The miRtargetlink database (https://ccb-web.cs.uni-saarland.de/mirtargetlink/) was used to determine potential target mRNAs for hsa-let-7i-5p and hsa-let-7b-5p. Appendix A presents 119 target genes for hsa-let-7i-5p and 158 target genes for hsa-let-7b-5p (Sheets 2 and 3, respectively).

Early human embryogenesis up to the blastocyst stage is characterized by a wavy change in the expression levels of certain groups of genes, which reflects the clearance of maternal mRNAs and the activation of the zygotic genome. Impressive and important data for understanding early embryogenesis obtained by Zhang P et al. [11] demonstrated the transcription dynamics at six developmental stages by applying microarray analysis. Since the expression pattern of protein-coding genes on day 4 a.f. (morula stage) does not differ from that on day 3 a.f. (8-cell stage) according to Yan L et al. [9], we considered it reasonable to compare the target genes of upregulated sncRNAs in the group of morula with high development potential (Appendix A, Sheet 1–3) with the following gene lists from the article by Zhang P [11]: (i) downregulated genes on Day 3 in comparison with Day 2 after fertilization (Appendix A, Sheet 4); (ii) downregulated genes on Day 3 in comparison with Day 5 after fertilization (blastocyst stage) (Appendix A, Sheet 5); and (iii) downregulated genes on Day 5 in comparison with Day 3 after fertilization (Appendix A, Sheet 6). The intersections between these gene lists are presented in Table 5. Among the genes downregulated on Day 3 a.f. compared to Day 2 a.f., being the time period of MZT, we found common gene-targets for hsa-let-7b-5p and hsa-let-7i-5p (highlighted in bold in Table 5) and gene-targets unique to hsa-let-7b-5p, hsa-let-7i-5p, hsa_piR_011291, hsa_piR_019122, hsa_piR_001311, hsa_piR_015026, hsa_piR_015462, hsa_piR_016735, hsa_piR_019675, hsa_piR_020381, hsa_piR_004880. In addition, the target genes of the sncRNAs indicated in Table 5 were genes, which expression level was reduced at the 8-cell stage relative to the blastocyst stage. Possibly, inhibition of the expression level of these genes, in part under the control of sncRNAs at the 8-cell stage, is necessary for the onset of blastulation and further embryo development. We also found sncRNA gene-targets, in particular, *TPD52, KIFC3, AGPAT3, LRRC17, XYLT1, WDR37, TRAFD1*, and *DAAM1*, in which there was a decrease in expression both at Day 3 vs. Day 2 and at Day 5 vs. Day 3, which may indicate a continual decrease in the expression of this set of genes in the period from the third to the fifth day after fertilization, presumably under the control of an sncRNA whose levels were identified to be increased in this work. It is important to underline that *BUB1B* gene-target down-regulated at Day 3 compared to Day 2, and the *MTAP* and *MLLT4* gene-targets downregulated at Day 3 compared to Day 5 are maternal genes that determine the quality of the oocyte and the development potential of the resulting embryo [39].

The molecular function of sncRNAs gene-targets downregulated at Day 3 vs. Day 2, Day 3 vs. Day 5, and Day 5 vs. Day 3 were analyzed using the Functional Enrichment analysis tool (http://www.funrich.org/), and the results are graphically represented in Figure 3a–c, respectively. 

## 3. Discussion

In vitro fertilization outcome depends on many factors such as patient age, genetics, reproductive system diseases, variability in the quality of oocytes and endometrium receptivity under the influence of exogenous gonadotropins, and sperm fertility [40]. In women with advanced maternal age, oocyte nucleus and cytoplasm maturation are impaired, which results in incorrect transition from maternal to embryonic genome activation during early embryo development [41]. This process leads to compromised competence of the resultant embryos. Advanced maternal age is one of the main causes of embryo development failure at the blastocyst stage and the development of blastocyst aneuploidy [42,43]. It has been shown that women over 35 years old have a higher rate of degraded and arrested embryos [43]. Multiple mitochondrial dysfunctions, shortening of telomeres, cohesin dysfunctions, and spindle instability are some of the negative consequences resulting from impairment of the main mechanisms related to aging, and their failure results in reduced oocyte and embryo competence [44,45,46,47]. The activities of gene products involved in cell cycle regulation are also altered in older women [48]. These data are in good agreement with the present study’s findings of statistically significant positive correlations between the patient’s age and the percentage of degenerated morula but inverse correlations with the number of OCC, number of oocytes at stage MII, and the serum level of AMH. AMH is one of the standard markers of ovarian reserve, oocyte and derived embryo quality, and euploidy. Gat et al. described a significant association between serum AMH and the proportion of euploid embryos [49]. Moreover, higher AMH has been found to be associated with improved rates of implantation, pregnancy, and live birth [50]. Impaired AMH expression among patients older than 37 years old may contribute to worse oocyte and embryo quality. It has been demonstrated that AMH and other proteins are involved in TGF-β signaling pathways, which are essential for competent follicular development and oocyte maturation [51]. In this regard, the choice of the ovarian stimulation protocol is individualized where, compared with r-FSH, HMG might be slightly more beneficial for patients with advanced maternal age or women with severe gonadotropin deficiency [52].

Despite advances in reproductive technology in recent years, the pregnancy rate per transferred embryo is still low [53]. To increase the probability of successful pregnancy, it is very important to select the embryo with the highest development potential. To date, the morphology of resultant embryos is considered the primary method utilized by embryologists to assess development and to select embryos for transfer into the uterine cavity. Gene expression profiling and analysis is emerging as a promising tool to improve the understanding and, thereby, assessment of what determines the quality of the embryo. Many researchers have attempted to characterize the expression profile of protein-coding genes at different stages of early embryogenesis [11,54,55]. SncRNAs are important regulators of their expression at the transcriptional and post-transcriptional levels. Sari I and colleagues [56] have concluded that an increase in the gonadotropin dose and frequency of their use leads to a decrease in the quality of oocytes and, as a consequence, to impaired embryogenesis, one of the reasons for which is a change in the protein expression profile involved in the biogenesis of piRNA.

Therefore, for a more complete understanding of the molecular mechanisms of regulation of early embryogenesis in the present study, we compared the overexpressed hsa_piR_011291, hsa_piR_019122, hsa_piR_001311, hsa_piR_015026, hsa_piR_015462, hsa_piR_016735, hsa_piR_019675, hsa_piR_020381, hsa_piR_004880, hsa-let-7b-5p, and hsa-let-7i-5p in spent culture medium of embryos at the morula stage with high developmental potential (good/excellent quality blastocyst capable of implanting and leading to the birth of a healthy child) with the expression profile of their potential target genes, the expression level of which, according to Zhang P et al. [11], was reduced at the stage of the maternal–zygotic transition until the blastocyst stage.

Identified target genes encode gene transcriptional regulators (TEAD3, TEAD1, SP1, HOXB6, ZNF557, DLX4, ZBTB38, ZNF814, PHF16, ELF1, GON4L, ZNF280B, SP3, and GBX2); metabolite interconversion enzymes, particularly glycosyltransferase, deacetylase, hydrolase, kinase, ligase, oxidoreductase, and phospholipase (MTAP, EXT2, MGLL, NAGA, ARG2, MGAT5B, SOD2, B3GALT6, PDXK, QARS, ACSL6, FOXRED1, B3GNT5, and CSGALNACT1); RNA-binding proteins (MEX3D, G3BP2, IGF2BP2, and RBPMS); protein-modifying enzymes (HEMK1, NEDD4L, PCSK6, AKT2, CNDP2, MID1, and BUB1B); transmembrane signal receptors (FZD5, GABBR2, and LRRC17); protein-binding activity modulators (NRAS, RGS16, GNAL, and ARHGAP28); scaffold/adaptor proteins (DEPDC1, MVP, GAB2, AKAP1, ELMOD2, and BTRC); transporters (ATP2B1, SLC45A4, and PLSCR3); and membrane trafficking proteins (MX1, VPS33A, REEP3, EHD4, SYN2). 

According to data in the literature, the gene expression levels of the discussed genes are associated with gamete maturation and pre-implantation development. In particular, the ESR1/SP1/CREBBP pathway was altered in embryos from women of advanced age [57]; TEAD activity is necessary for inner cell mass formation quality, which is important for proper embryo development [58], and it was found that the YAP/TEAD3 signaling pathway was implicated in trophoblast implantation to the maternal endometrium [59]; GON4L coordinates morphogenesis along the anteroposterior embryonic axis [60]; IGF2BP2 is a critical maternal-derived factor that participates in early zygotic genome activation, and its maternal deletion in mouse embryos causes early in vitro embryonic developmental arrest at the 2-cell stage [61]; Akt2 is necessary for normal embryo progression through cleavage stages and is involved in blastulation [62]; BUB1B is involved in chromosome segregation and ploidy status regulation in oocytes and is associated with developmental potential of human pre-implantation zygotes [63]. Confirmation of the functional significance of this group of target genes in the field of reproduction by various research teams underlines the importance of the data obtained in the present study for the quantitative assessment of sncRNAs that are potentially regulating their expression levels.

It should be noted that hsa_piR_015026, hsa_piR_020485 and hsa_piR_000807 were ignored not incidentally when searching for gene-targets of sncRNAs implicated in MZT and blastulation. The reason is their reduced detection level in the spent culture medium from morula with no development potential compared to the reference culture medium unexposed to embryos. We did not consider these piRNAs because we do not fully understand the real cause for this decrease in the detection level. This may be due to the increased uptake of these piRNAs by the morula from the culture medium. It is unclear whether this is a cause or a consequence of anomaly in morula development. However, it is important to understand the possible influence of sncRNAs included in the culture media used on MZT and embryonic genome activation. In addition, the quantitative and qualitative composition of sncRNA in the culture medium from different manufacturers may vary, which may affect the effectiveness of ART programs and the interpretation of the data obtained.

In summary, we have presented new original data on small RNA sequencing of spent culture media from morula of different development potential with respect to implantation success. Our findings provide a view of a more complex network that controls human embryogenesis at the pre-implantation stage. Further research is needed using reporter analysis to experimentally confirm interactions between identified sncRNA/gene target pairs.

## 4. Materials and Methods

### 4.1. Experimental Design

The research consisted of three main stages: (1) embryological stage with the formation of morula groups with different development potential; (2) quantitative analysis of sncRNA in morula groups by deep sequencing and qPCR; (3) search for target genes of sncRNAs involved in MZT and blastulation (Figure 4).

Written informed consent was obtained from each patient and the study was approved by the ethics committee of the National Medical Research Center for Obstetrics, Gynecology, and Perinatology, named after Academician V.I. Kulakov of Ministry of Healthcare of the Russian Federation (protocol No 6, approval date: 29 August 2019).

### 4.2. Clinical Characteristics of Couples and Stimulation Protocol in the ART Program

In the current study, 27 subfertile couples were included (Appendix A). The average female age was 33 years, while the average male age was 35 years. Women’s average body mass index (BMI) was 23.4 kg/m^2^. All men included in this study underwent spermogram analysis, which was performed on the day of transvaginal ovarian puncture. From this analysis, 18 patients were found to have teratozoospermia; 4, oligoteratozoospermia; 2, oligoasthenoteratozoospermia; 2, asthenoteratozoospermia; and 1, normozoospermia. In 10 out of 27 assessed couples, a combination of tubal and male factor infertility was diagnosed. Two patients also had small uterine myomas which did not require any surgery, and 4 patients had diminished ovarian reserve (DOR).

Gonadotropin administration in a protocol with ant-GnRH on day 2–3 of the menstrual cycle was included in the current study for all patients. In 13 patients, recombinant follicle-stimulating hormone (r-FSH) was used, and in 8 women, human menopausal gonadotropins (HMGs) were administered. One patient underwent an injection of the r-FSH depo form, and 3 patients had both r-FSH and HMG. For follicular development synchronization, ant-GnRH was prescribed for 2 days at a dose of 0.25 mg/day subcutaneously in one case on the second day of the menstrual cycle, followed by the administration of gonadotropins (HMG + r-FSH) according to the standard scheme. In one patient, a fresh donor oocyte IVF/ICSI cycle was performed due to the DOR. The endometrium was prepared by estradiol from the 5th day of the menstrual cycle.

When the follicle reached 14 mm in diameter, ant-GnRH at a dose of 0.25 mg/day was applicated subcutaneously for 4–6 days to prevent a premature LH surge. To induct final oocyte maturation after the follicles reached ≥17 mm in diameter, human chorionic gonadotropin (hcG) was administered at a dose of 10,000 IU in 18 patients, at a dose of 9000 IU in 3 patients, and at a dose of 8000 IU in 1 woman. To prevent ovarian hyperstimulation syndrome (OHSS) in 4 patients, 0.2 mg decapeptyl was used as an alternative for final maturation induction. The average dose of gonadotropins in the patients included in the study was 1885 UI, and ovarian stimulation lasted approximately 10 days. Collected oocytes were fertilized by ICSI method. Couples with contraindications for the IVF/ICSI program were not included in the study.

### 4.3. Oocytes Fertilization Protocol

Immediately after follicular fluid aspiration during oocyte retrieval, the number of oocyte–cumulus complexes (OCC) and the maturity of the retrieved oocytes were identified under a stereomicroscope on the heated surface of a sterile laminar box. A stable temperature (37.0 °C) was constantly maintained during all manipulations. For pre-incubation, all OCC were washed from follicular fluid and blood and placed in sterile plates (Thermo Fisher Scientific Nunc A/S, Denmark, Roskilde) with Continuous Single Culture medium (CSCM, Irvine Sc., USA, CA, Santa Ana) for 2–3 h at a temperature of 37.0 °C and with 6% CO_2_. After the pre-incubation period, the oocytes were denuded and cumulus cells that surrounded the oocytes were removed using hyaluronidase solution (Irvine Sc., USA, CA, Santa Ana). The oocytes were placed in hyaluronidase solution for 2 min. 

Then, OCCs were washed again in the CSCM and returned to the wells. Retrieved oocytes at metaphase II stage were fertilized by the ICSI method and then transferred back to the CSCM for further cultivation. 

ICSI was performed next in the following stages:

1. Oocyte preparation for spermatozoa injection (removal of cumulus cells); 2. spermatozoa selection; 3. spermatozoa aspiration into the injection pipette; 4. oocyte fixation and rotation to localize the polar body; 5. injection of the spermatozoa into the oocyte.

The appearance of two pronuclei was observed 14–16 h after fertilization. If the presence of two pronuclei in the oocyte was not visualized at that point, the fertilization was considered failed. All embryos were cultivated in multigas incubators, produced by COOK (Australia, Brisbane), in 25 μL drops with oil (Fujifilm, Irvine Sc., USA, CA, Santa Ana). CSCM-C (Continuous Single Culture Complete) was not changed during 3 days of embryo cultivation. On the 4th day after fertilization, all embryos were graded by their morphological characteristics according to Tao J. et al. [64]. The embryo was considered to be at the morula stage provided the primary cavity had not yet formed. At the same time, the boundaries between cells in morula were barely detectable or not visualized (indicating a compact morula-stage embryo). On day 4 after fertilization, embryos were transferred into 25 μL of fresh CSCM-C medium for further cultivation until the 5th day after fertilization. A 25 μL aliquot of spent culture medium was collected from each embryo on day 4 after fertilization into individual sterile tubes (SSI, USA, CA, Lodi) and subsequently frozen in liquid nitrogen and stored at −70 °C for sncRNA expression profile analysis by deep sequencing and qPCR.

The embryos preferred for transfer were excellent/good-quality blastocysts, but fair quality blastocysts were transferred if there were no better alternatives. If blastocysts were routinely of bad quality, they were not recommended for uterine transfer and the IVF cycle was interrupted. 

Depending on the outcome of morula development on the 5th day after fertilization, 4 groups of samples of the spent culture medium were formed and collected on the 4th day after fertilization:

(I) morula with high potential for blastulation (development into a blastocyst of good or excellent quality on the 5th day after fertilization according to Istanbul Consensus Workshop [65])—12 blastocysts on the 5th day were transferred into the uterine cavity followed by the birth of a healthy child; (II) morula with a low potential for blastulation (development into a blastocyst of poor/fair quality on the 5th day after fertilization according to Istanbul Consensus Workshop [65])—8 samples; (III) morula without the potential for blastulation (degenerated by the 5th day after fertilization)—20 samples; (IV) morula arrested at the morula stage on the 5th day after fertilization—9 samples.

### 4.4. Extraction of RNA from Spent Culture Medium

Twenty-five microliters of embryo culture medium adjusted to 200 μL with 0.9% NaCl were treated with 1000 µL of QIAzol Lysis Reagent (Qiagen, Hilden, Germany) followed by mixing with 200 µL of chloroform, centrifugation for 15 min at 12,000× *g* (4 °C), collection of 600 µL aqueous phase, and RNA isolation using the miRNeasy Serum/Plasma Kit (Qiagen, Hilden, Germany).

### 4.5. cDNA Library Preparation and RNA Deep Sequencing

cDNA libraries were synthesized using 7 µL of the 14 µL total RNA column eluate (miRNeasy Serum/Plasma Kit, Qiagen, Hilden, Germany), extracted from spent culture medium using the NEBNext^®^ Multiplex Small RNA Library Prep Set for Illumina^®^ (Set1 and Set2, New England Biolab^®^, Frankfurt am Main, Germany), amplified for 30 PCR cycles, and sequenced on the NextSeq 500 platform (Illumina, San Diego, CA, USA). The adapters were removed using Cutadapt. All trimmed reads shorter than 16 bp and longer than 50 bp were filtered out. The remaining reads were mapped to the GRCh38.p15 human genomes, miRBase v21, and piRNABase with bowtie aligner [66]. Aligned reads were counted using the featureCount tool from the Subread package [67] and with the fracOverlap 0.9 option, so the whole read was forced to have 90% intersection with sncRNA features. Differential expression analysis of the sncRNA count data was performed with the DESeq2 package [68].

### 4.6. Quantitative Real-Time RT-PCR 

Five microliters of the 14 µL total RNA column eluate (miRNeasy Serum/Plasma Kit, Qiagen, Hilden, Germany) extracted from the embryo culture medium was converted into cDNA in a reaction mixture (20 µL) containing 1× HiSpec buffer, 1× Nucleics mix, and miScript RT, according to the miScript^®^ II RT Kit protocol (Qiagen, Hilden, Germany); then, the sample volume was adjusted with deionized water to 200 µL. The synthesized cDNA (2 µL) was used as a template for qPCR using a forward primer specific to the studied sncRNA (Table 3) and the miScript SYBR Green PCR Kit (Qiagen, Hilden, Germany). The following qPCR conditions were used: (1) 15 min at 95 °C and (2) 50 cycles at 94 °C for 15 s, an optimized annealing temperature (45–61.6 °C) for 30 s, and 70 °C for 30 s; followed by heating the reaction mixture from 65 to 95 °C in 0.1 °C increments to plot the melting curve of the qPCR product in a StepOnePlus™ thermocycler (Applied Biosystems, Foster City, CA, USA). The relative expression of sncRNA in the embryo culture medium was determined using the ∆∆Ct method using hsa_piR_023338 (DQ601914, GenBank, available online: https://www.ncbi.nlm.nih.gov/genbank/) as the reference RNA and culture medium without contact with any embryo incubated for 4 days at 37 °C as a reference sample to calculate the fold change of expression level in a sample. hsa_piR_023338 was chosen as the reference RNA due to its consistent expression level in all 51 analyzed samples.

### 4.7. Statistical Analysis of the Obtained Data

For statistical processing of the results, we used scripts written in the R language [67] and RStudio [69]. The correspondence of the analyzed parameters to the normal distribution law was assessed using the Shapiro–Wilk test. When the distribution of data was different from normal, the Mann–Whitney test was used for paired comparison, and data were described as median (Me) and quartiles Q1 and Q3 in the Me format (Q1; Q3). To identify the relationship between categorical variables, chi-square testing was performed. Since both quantitative and qualitative characteristics were analyzed, the correlation analysis was performed using Spearman’s nonparametric correlation test. The 95% confidence interval for the correlation coefficient was determined using Fisher transformation. The value of the threshold significance level *p* was taken as equal to 0.05. If the *p*-value was less than 0.001, then it was indicated in the format *p* < 0.001.

### 4.8. Ethics Statement

The ethics committee of the National Medical Research Center for Obstetrics, Gynecology, and Perinatology, named after Academician V.I. Kulakov of Ministry of Healthcare of the Russian Federation, approved this study (ethics committee approval protocol No 6, approval date: 29 August 2019).

## Figures and Tables

**Figure 1 ijms-21-09399-f001:**
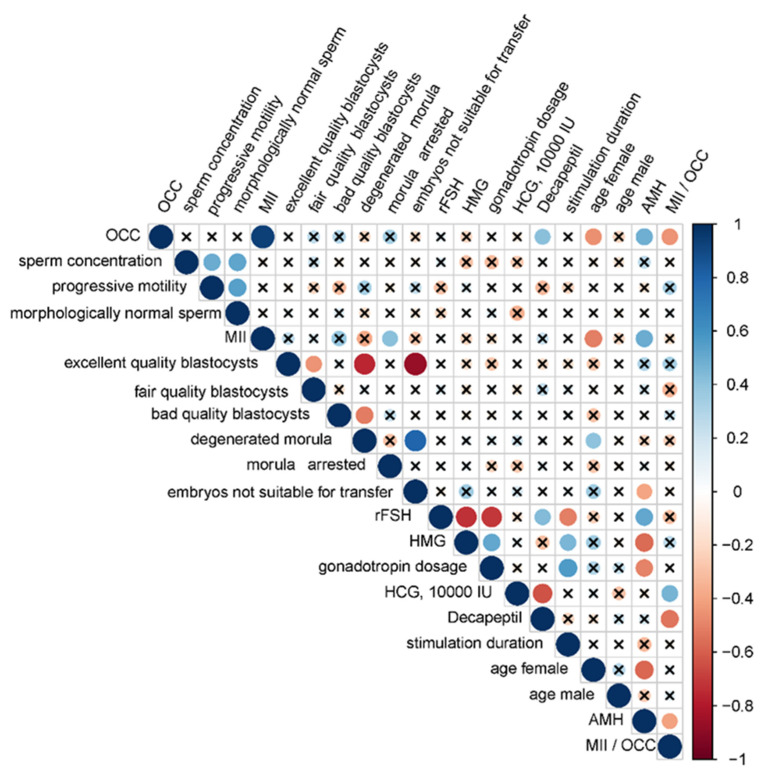
Correlation matrix based on the nonparametric Spearman rank correlation method. Significant (*p* < 0.05) correlations are indicated by a dot, nonsignificant correlations are indicated by a cross, positive correlations are marked in blue, and negative correlations in red: the more significant the correlation, the larger the dot size. Parameters used in correlation analysis: OCC—the number of oocyte–cumulus complexes from the female of each couple; sperm concentration—spermatozoid count per milliliter of ejaculate from the male of each couple; progressive motility—percentage of linearly motile spermatozoids from the male of each couple; morphologically normal sperm—percentage of morphologically normal spermatozoids from the male of each couple; MII—metaphase II oocyte number from the female of each couple; excellent quality blastocysts—percentage of excellent/good-quality blastocysts on day 5 a.f.; fair quality blastocysts—percentage of fair-quality blastocysts on day 5 a.f.; bad quality blastocysts—percentage of bad-quality blastocysts on day 5 a.f.; degenerated morula—percentage of degraded morula on day 5 a.f.; morula arrested—percentage of morula arrested in development on day 5 a.f.; embryos not suitable for transfer—percentage of degraded morula, morula arrested in development and morula developed into a blastocyst of bad quality on day 5 a.f.; rFSH—the use of recombinant follicle-stimulating hormone for ovarian stimulation; HMG—the use of human menopausal gonadotropins for ovarian stimulation; gonadotropin dosage —total dose of gonadotropin used for ovarian stimulation, IU; HCG, 10,000 IU—the use of human chorionic gonadotropin (10,000 IU) for triggering final oocyte maturation; Decapeptil—the use of 0.2 mg decapeptyl for triggering final oocyte maturation; stimulation duration—duration of ovarian stimulation, days; AMH—the anti-Müllerian hormone level (ng/mL) in blood from the female of each couple; MII/OCC—metaphase II oocyte number as a percentage of oocyte–cumulus complexes.

**Figure 2 ijms-21-09399-f002:**
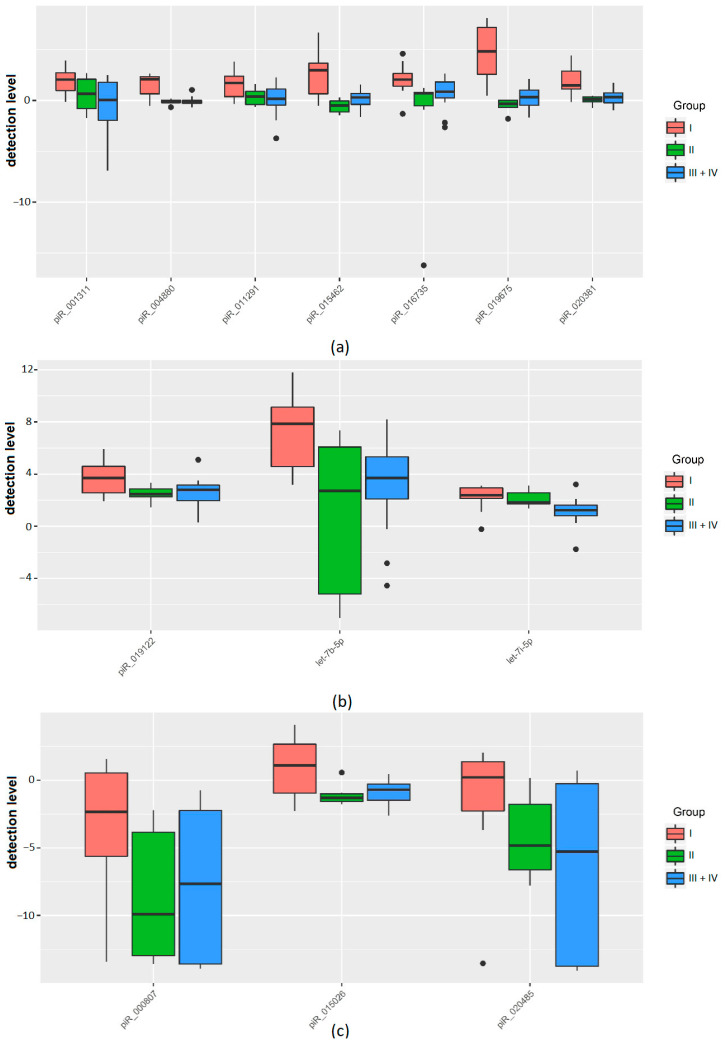
Box plots of the qPCR data in the form of base 2 logarithm of the fold change in the detection level of sncRNA in the groups of samples I–IV relative to the reference sample. (**a**) An increase in the sncRNA detection level fold change in group I and a slight increase (less than 2 times) or no fold change in sncRNA detection level in groups II–IV. (**b**) An increase in the sncRNA detection level fold change in all groups by more than 2 times with more pronounced changes in group I. (**c**) A decrease in the sncRNA detection level fold change in the studied groups. The statistical significance of the differences between the groups is presented in Table 4.

**Figure 3 ijms-21-09399-f003:**
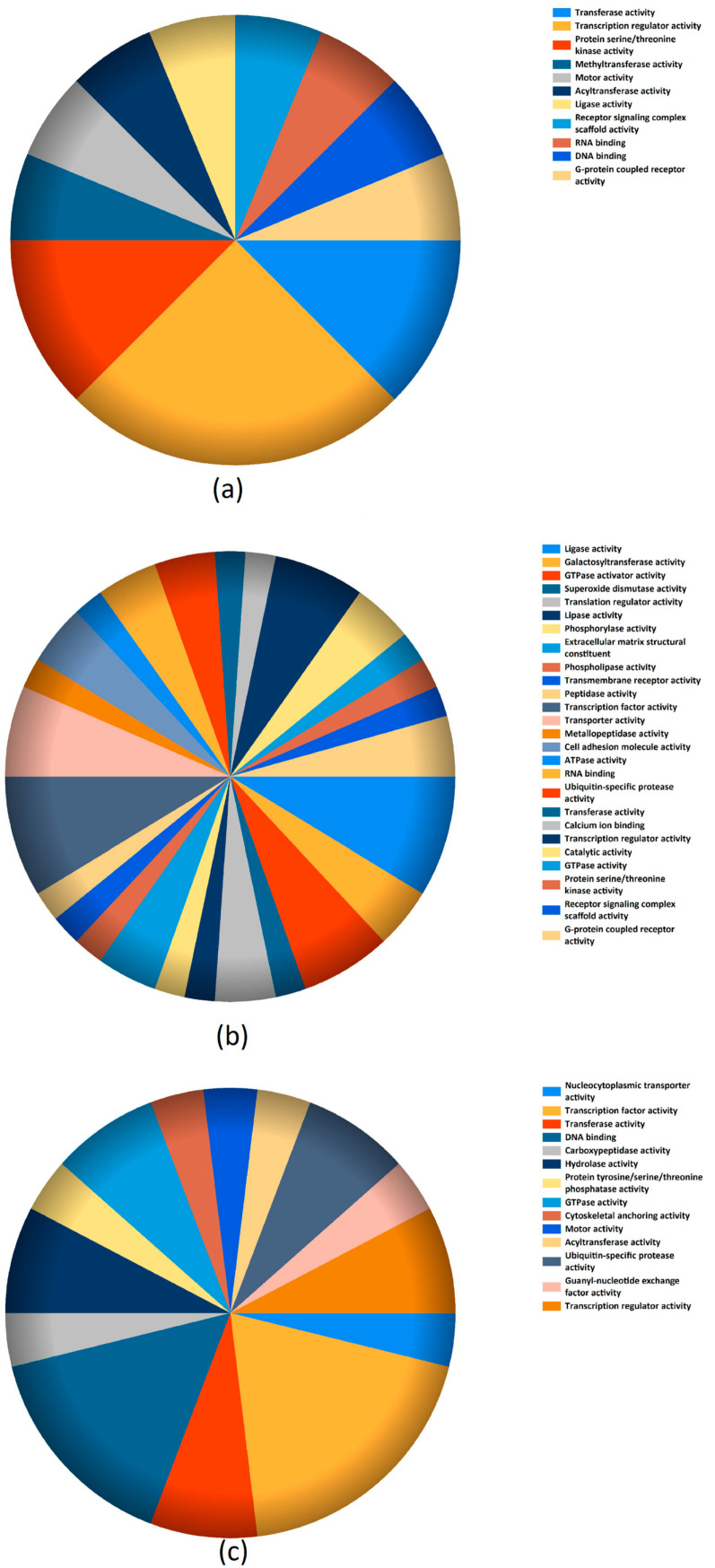
Molecular function of sncRNAs gene targets downregulated at Day 3 vs. Day 2 (**a**), Day 3 vs. Day 5 (**b**), and Day 5 vs. Day 3 (**c**).

**Figure 4 ijms-21-09399-f004:**
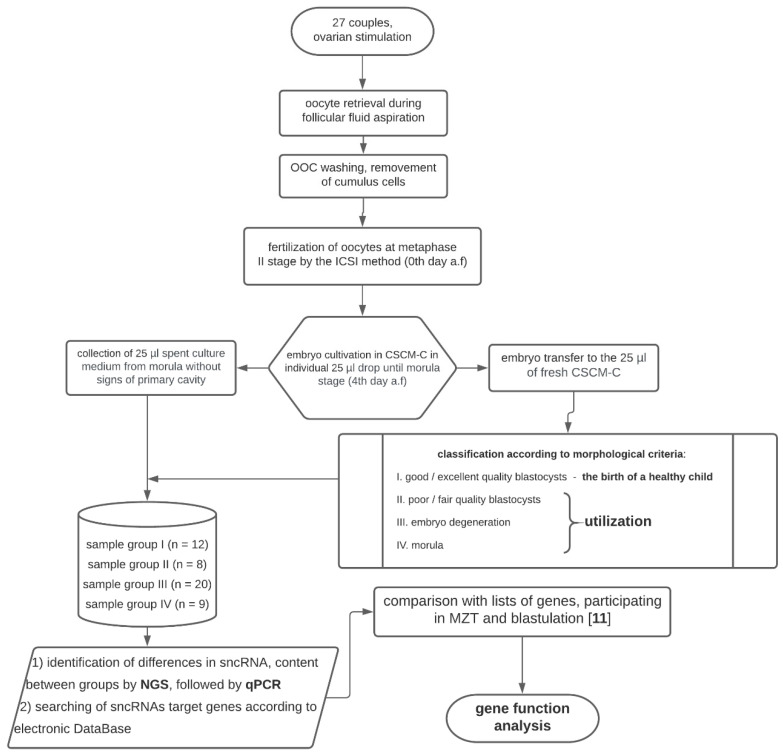
Flow diagram of the experimental design.

**Table 1 ijms-21-09399-t001:** Comparative characteristics of clinical data and parameters of the IVF/ICSI protocol in couples depending on the outcome of the ART program.

**IVF/ICSI Protocol Result**	**Age of Female ^1^**	**Age of Male ^1^**	**Female, BMI ^1^**	**Female, AMH, ng/mL ^1^**	**Number of OCC ^1^**	**Number of Metaphase II (MII) Oocytes ^1^**	**Sperm Concentration, Million per Milliliter ^1^**	**Sperm with Progressive Motility, % ^1^**	**Morphologically Normal Spermatozoa, % ^1^**	**r-FSH for Ovarian Stimulation ^2^**
delivery (*n* = 11)	30 (29.5, 32)	32 (31.5, 34)	23 (23, 24)	2.7 (2.15, 3.1)	10 (4.5, 13)	6 (4, 10)	84 (36, 90)	65 (39.5, 70)	2 (1.5, 3)	9 (81, 8%)
negative (*n* = 16)	33 (30, 38)	35 (33, 36)	24 (22.75, 24)	1.6 (1.17, 2.13)	10 (4, 14.5)	6 (3.5, 8)	57 (26.75, 73)	65 (39.5, 70)	2.5 (1.75, 3)	4 (25%)
*p*	0.0702	0.1713	0.9581	0.0061	0.9792	0.7149	0.2667	0.7669	0.7322	0.0065
**IVF/ICSI Protocol Result**	**HMG for Ovarian Stimulation ^2^**	**Gonadotropin Dosage ^1^**	**Stimulation Duration, Days ^1^**	**HCG for Triggering Final Oocyte Maturation, 8000–10,000 IU ^2^**	**Decapeptyl for Triggering Final Oocyte Maturation, 0.2 mg ^2^**	**% Excellent/Good-Quality Blastocysts on Day 5 a.f. ^1^**	**% Fair-Quality Blastocysts on Day 5 a.f. ^1^**	**% Bad-Quality Blastocysts on Day 5 a.f. ^1^**	**% Degraded Morula on Day 5 a.f. ^1^**	**% Morula Arrested in Development on Day 5 a.f. ^1^**
delivery (*n* = 11)	0 (0%)	1650 (1200, 1987.5)	9 (8.5, 10)	10 (90.9%)	1 (9.1%)	50 (24, 58)	0 (0, 12.5)	0 (0, 8.5)	50 (16.5, 50)	0 (0, 5.5)
negative (*n* = 16)	9 (56.3%)	2100 (1850, 2550)	11 (9.5, 11)	12 (75%)	3 (18.8%)	35.5 (0, 67.5)	0 (0, 0)	0 (0, 0)	29.5 (0, 54.25)	0 (0, 22.75)
*p*	0.0025	0.0101	0.0226	0.3833	0.6252	0.822	0.3697	0.4232	0.7997	0.4272

^1^ data are presented as a median (Me) and quartiles Q1 and Q3 in the Me format (Q1; Q3) with an indication of the statistical significance (*p*) using the Mann–Whitney test; ^2^ data are presented as absolute numbers N and percentages of the total number of patients in a group P in the format N (P%) with an indication of the statistical significance (*p*) while using the χ^2^ test; IVF-in vitro fertilization; ICSI-intracytoplasmic sperm injection; ART-assisted reproductive technology; r-FSH-recombinant follicle stimulating hormone; AMH-anti-Müllerian hormone; OCC-oocyte–cumulus complexes; HCG-human chorionic gonadotropin.

**Table 2 ijms-21-09399-t002:** Small noncoding RNA deep sequencing data.

sncRNA	Sample Group (I-IV), Sample ID (#) ^1^														I vs. (III + IV), *p*-Value	II vs. (III + IV), *p*-Value
	I	I	I	II	II	II	III	III	III	III	IV	IV	IV	Reference		
	#150	#118	#154	#170	#176	#147	#124	#157	#206	#200	#127	#177	#134	#207		
**hsa_piR_011291**	69	80	58	40	29	27	26	4	69	53	24	42	17	55	0.017	0.004
**hsa_piR_019122**	42	23	54	31	5	7	19	8	11	9	7	12	5	35	0.001	0.054
**hsa_piR_001311**	150	66	150	207	15	78	96	21	87	81	54	87	75	48	0.029	0.372
**hsa_piR_015026**	120	65	140	185	15	55	75	10	35	70	80	110	45	15	0.041	0.349
**hsa_piR_015462**	1290	1302	1688	1156	392	884	1764	428	1046	668	700	1068	594	1102	0.047	0.038
**hsa_piR_016735**	170	232	265	174	106	207	221	35	155	176	85	178	40	104	0.039	0.107
**hsa_piR_019675**	2563	2074	4699	1865	947	1639	2913	661	1941	1823	1185	1392	663	1847	0.023	0.064
**hsa_piR_020381**	9635	7413	14,019	6381	5229	7449	5778	1965	6726	6621	3318	4833	1497	5532	0.004	0.061
**hsa_piR_020485**	170	250	340	1078	147	525	672	182	672	448	343	357	560	532	0.052	0.148
**hsa_piR_004880**	754	651	839	568	195	439	878	213	518	333	347	532	297	549	0.029	0.023
**hsa_piR_000807**	1290	1300	1684	1156	392	880	1760	428	1046	668	700	1064	594	1102	0.046	0.038
**hsa_piR_001312**	450	198	450	621	54	234	288	63	261	252	162	261	234	144	0.031	0.377
**hsa_piR_020365**	56	53	40	97	19	13	46	11	53	47	8	11	14	9	0.055	0.410
**hsa_piR_022628**	28	119	133	28	42	175	21	7	21	28	0	14	28	14	0.003	0.424
**hsa_piR_022104**	200	314	471	628	0	0	471	0	0	157	157	0	0	1099	0.048	0.311
**hsa_piR_019752**	135	243	45	90	9	108	18	9	108	81	36	45	18	72	0.023	0.165
**hsa**_piR_019269	33	23	55	31	5	7	19	8	11	9	7	12	5	35	0.001	0.073
**hsa**_piR_006927	32	23	54	31	5	7	19	8	11	9	7	12	5	35	0.001	0.076
**hsa**_piR_002769	10	82	164	41	0	0	0	0	0	0	0	0	0	0	0.006	0.099
**hsa**_piR_008112	14	7	9	0	0	0	1	0	14	7	0	0	0	0	0.042	0.004
**hsa**_piR_006710	18	16	24	16	2	4	14	0	4	4	14	10	2	20	0.005	0.037
**hsa**_piR_010119	35	78	83	43	21	56	41	30	35	64	17	24	36	29	0.023	0.120
**hsa**_piR_020668	30	53	192	84	17	24	25	6	41	28	26	45	25	71	0.038	0.207
**hsa**_piR_018552	17	6	18	11	2	7	12	3	10	7	7	5	5	11	0.027	0.103
**hsa-let_7b_5p**	35	29	32	41	11	9	34	9	6	2	20	9	32	17	0.036	0.164
**hsa-let_7i_5p**	29	23	26	10	26	10	24	0	1	1	14	14	30	18	0.045	0.065

^1^ data are presented as sncRNAs read counts in the following sample groups: I—morula developed into a blastocyst of good or excellent quality on day 5 a.f. and transferred into the uterine cavity followed by the birth of a healthy child; II—morula developed into a blastocyst of poor/fair quality on day 5 a.f.; III—morula degenerated by day 5 a.f.; IV—morula arrested at the morula stage on day 5 a.f.; Reference—culture medium without contact with any embryo. Sample ID is specified in Arabic numerals. *p*—statistical significance of the differences between the groups.

**Table 3 ijms-21-09399-t003:** sncRNA sequence data.

sncRNA ^1^	Accession Number ^1^	Nucleotide Sequence of Sense Primer for PCR, 5′-3′	PCR Primers Annealing Temperature, °C
hsa_piR_011291	DQ585247	TGCGACTCACTGTAGTGCTGGGGATCC	46.2
hsa_piR_019122	DQ596252	GACAGAGAAAACAAGGTGGTGAACTATGCCC	46.2
hsa_piR_001311	DQ571812	ATTGGTGGTTCAGTGGTAGAATTCTCGCC	45
hsa_piR_015026	DQ590548	TGGTTCAGTGGTAGAATTCTCGCCTCC	45
hsa_piR_015462	DQ591122	CCTGGGCCAGCCTGATGATGTCCTCCTC	45
hsa_piR_016735	DQ593039	CCTGGGAATACCGGGTGCTGTAGGCTTA	50
hsa_piR_019675	DQ596992	GCAATAACAGGTCTGTGATGCCCTTAGA	53
hsa_piR_020381	DQ597997	GGCGGGAGTAACTATGACTCTCTTAAGGTA	53
hsa_piR_020485	DQ598159	GATGTAGCTCAGTGGTAGAGCGCATGCT	53
hsa_piR_004880	DQ576715	TTGTCCTGGACCAGCCTGATGATGTCCTC	45
hsa_piR_000807	DQ571005	CTGATGATGTCCTCCTCCAGTTGCCGC	53
hsa_piR_001312	DQ571813	ATTGGTGGTTCAGTGGTAGAATTCTCGCCTG	46.2
hsa_piR_020365	DQ597975	GGCCGTGATCGTATAGTGGTTAGTACTCTG	46.2
hsa_piR_022628	DQ600952	TAGAGCATGAGACTCTTAATCTCAGGGTCGTG	48.9
hsa_piR_022104	DQ600278	TACCTAGGTGATGGGATGATCTGTGC	48.9
hsa_piR_020388	DQ598008	GGCTCGTTGGTCTAGGGGTATGATTCTCGG	45
hsa_piR_019752	DQ597110	GCAGAGTGGCGCAGCGGAAGCGTGCTGGGCCC	61.6
hsa-let-7b-5p	MIMAT0000063	Hs_let-7b_1 miScript Primer Assay, Cat.No. MS00003122	55
hsa-let-7i-5p	MIMAT0000415	Hs_let-7i_1 miScript Primer Assay, Cat.No. MS00003157	55

^1^ piRNAbank (http://pirnabank.ibab.ac.in/cgi-bin/accession.cgi) for piRNAs; miRBase (http://www.mirbase.org/search.shtml) for miRNAs.

**Table 4 ijms-21-09399-t004:** Pairwise comparison of morula groups (I–IV) with different developmental potential according to the fold change in the detection level of sncRNAs in spent culture medium assessed by qPCR.

sncRNA	Sample Group	Me ^1^	Log_2_(Me)	Log_2_(Q1)	Log_2_(Q3)	*p*-Value		
						I vs. III + IV	I vs. II	II vs. III + IV
hsa_piR_011291	I	3.3173	1.73	0.38	2.38	0.023651		
	III + IV	1.6245	0.7	−0.31	1.14			
	II	1.3104	0.39	−0.4	0.9			
hsa_piR_019122	I	13.0864	3.71	2.56	4.6	0.043099		
	III + IV	6.9163	2.79	2.11	3.31			
	II	5.5022	2.46	2.26	2.87			
hsa_piR_001311	I	4.1699	2.06	0.98	2.73	0.003247		
	III + IV	0.9862	−0.02	−1.86	1.09			
	II	1.5801	0.66	−0.79	2.09			
hsa_piR_015026	I	2.1435	1.1	−0.95	2.67	0.030825		
	III + IV	0.5987	−0.74	−1.47	−0.08			
	II	0.4033	−1.31	−1.56	−1.01			
hsa_piR_015462	I	7.9447	2.99	0.65	3.67	0.003247	0.015984	
	III + IV	1.0210	0.03	−0.58	0.62			
	II	0.7022	−0.51	−1.14	−0.04			
hsa_piR_016735	I	4.1699	2.06	1.41	2.66	0.013416		
	III + IV	1.8404	0.88	0.3	1.58			
	II	1.6358	0.71	−0.54	0.81			
hsa_piR_019675	I	28.2465	4.82	2.58	7.2	*p* < 0.001	*p* < 0.001	0.031124
	III + IV	1.2834	0.36	−0.33	1.01			
	II	0.7900	−0.34	−0.67	0.01			
hsa_piR_020381	I	2.8089	1.49	1.14	2.88	0.003247	0.002997	
	III + IV	1.3104	0.39	0.04	1.26			
	II	1.0792	0.11	−0.12	0.35			
hsa_piR_020485	I	1.1567	0.21	−2.27	1.37	0.001523		
	III + IV	0.0171	−5.87	−13.71	−2.95			
	II	0.0349	−4.84	−6.64	−1.78			
hsa_piR_004880	I	4.2871	2.1	0.65	2.34	*p* < 0.001	0.004745	
	III + IV	0.9266	−0.11	−0.36	0.09			
	II	0.9659	−0.05	−0.21	0.04			
hsa_piR_000807	I	0.1975	−2.34	−5.62	0.53	0.012145		
	III + IV	0.0013	−9.55	−13.58	−2.43			
	II	0.0011	−9.89	−12.96	−3.85			
hsa-let-7b-5p	I	230.7201	7.85	4.59	9.13	0.00462		
	III + IV	13.6422	3.77	1.58	5.26			
	II	6.6346	2.73	−5.2	6.07			
hsa-let-7i-5p	I	5.2416	2.39	2.15	2.95	0.001976		0.006596
	III + IV	2.1735	1.12	0.79	1.49			
	II	3.5554	1.83	1.72	2.58			

^1^ Data are presented as medians (Me) of the fold change in the detection level of sncRNA in the spent culture medium relative to the reference medium without any embryo and quartiles Q1 and Q3.

**Table 5 ijms-21-09399-t005:** Lists of genes with decreased expression level during MZT and blastulation which are possible targets of sncRNAs upregulated in the morula group (group I) with high potential for blastulation and implantation.

Stages of Embryo Development Being Compared	Target-Genes ^1^ of hsa-let-7b-5p (Appendix A, Sheet 3)	Target-Genes ^1^ of hsa-let-7i-5p (Appendix A, Sheet 2)	Target-Genes ^1^ of hsa_piR_011291, hsa_piR_019122, hsa_piR_001311, hsa_piR_015026, hsa_piR_015462, hsa_piR_016735, hsa_piR_019675, hsa_piR_020381, hsa_piR_004880 (Appendix A, Sheet 1)
8-cell stage (Day 3 a.f.) versus 4-cell stage (Day 2 a.f.), down-regulated genes are listed in Appendix A (Sheet 4)	*AGPAT3, **AKT2,** GAB2, GABBR2, LRRC17, MID1, RTTN, SPATA6, TAF9B, TCEB3B, **WDR37, XYLT1***	***AKT2,** BUB1B, DAAM1, TRAFD1, **WDR37, XYLT1***	*CSGALNACT1, EHMT1, KIFC3, LYPD6, MEX3D, TPD52, ZBTB38*
8-cell stage (Day 3 a.f.) versus blastocyst stage (Day 5 a.f.), down-regulated genes are listed in Appendix A (Sheet 5)	*ACSL6, **AKT2, ARHGAP28, ATP2B1**, ATPAF1, B3GNT5, CCR7, **FRAS1**, GAB2, IGF2BP2, IL11RA, KIAA0319L, MID1, MLLT4, PAPOLG, PCSK6, PDXK, PFAS, **PLCXD1**, PLSCR3, REEP3, RGS16, **VPS33A**, ZNRF1*	***AKT2, ARHGAP28, ATP2B1,** DEPDC1, DPH1, ELMOD2, FBXO22, **FRAS1,** FZD5, GPR56, IGSF3, MGLL, MTAP, NAGA, PHF16, **PLCXD1,** QARS, RBPMS, TEAD1, **VPS33A***	*B3GALT6, CNDP2, COL4A1, DLX4, EHD4, EXT2, FOXRED1, HEMK1, JUP, MX1, RGS3, SLC45A4, SOD2, SP3, STX3, TEAD3, TRERF1, VPS13A, ZBTB38*
Blastocyst stage (Day 5 a.f.) versus 8-cell stage (Day 3 a.f.), down-regulated genes are listed in Appendix A (Sheet 6)	*AGPAT3, ATPAF1, GNAL, **HOXA1**, LRRC17, NEDD4L, PAPOLG, **TMEM2**, **WDR37, XYLT1, ZNF280B,** ZNF324, **ZNF557, ZNF814***	*ARG2, CDC14B, CPD, DAAM1, **HOXA1,** NRAS, PAFAH1B2, PSD3, **TMEM2,** TRAFD1, **WDR37, XYLT1, ZNF280B, ZNF557, ZNF814***	*AKAP1, API5, BTRC, ELF1, G3BP2, GBX2, GON4L, HOXB6, KIFC3, MGAT5B, MVP, NPHP4, PAX8, PRAME, SP1, STX3, SYN2, TPD52*

^1^ the abbreviations of genes are defined according to the GeneCards database (https://www.genecards.org/) and presented in the section “Abbreviations”. The common gene-targets for hsa-let-7b-5p and hsa-let-7i-5p are highlighted in bold.

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
