# Peer review of "Small Noncoding RNA Signatures for Determining the Developmental Potential of an Embryo at the Morula Stage"

_ijms, 2020, doi:10.3390/ijms21249399_

Round 1
Reviewer 1 Report
The work by Timofeeva and co-authors studies the relationship between the levels of small non-coding RNAs (sncRNA) in the spent media of human embryos, the quality of the morula and the outcome of the assisted reproductive treatment. This study is a logical continuation for the work of the group, in particular, for their recent publication Timofeeva et al., 2019 (DOI: 10.3390/ijms20122912). In the present study, Timofeeva and co-authors show interesting data of sncRNA profiling from the culture media of human morulae of different qualities and developmental potentials. This study appears to be carefully conducted and the results are interesting. However, there are some issues that the authors have to address before I can recommend it for publication.Major comment: This reviewer is concerned about the low level of citation of existing literature on the field of miRNA analysis in embryo culture media as a predicting tool for pregnancy. Just to list some examples: Kropp et al., 2014 (DOI: 10.3389/fgene.2014.00091), Cuman et al., 2015 (DOI: 10.1016/j.ebiom.2015.09.003), Sánchez-Ribas et al., 2019 (DOI: 10.1177/1933719118766252), Timofeeva et al., 2019 (DOI: 10.3390/ijms20122912, from this same group), Kirkegaard et al., 2020 (DOI: 10.1007/s10815-020-01891-7), Abu-Halima et al., 2020 (DOI: 10.1016/j.fertnstert.2019.12.028). Citing and discussing the existing literature, comparing similarities and differences regarding methods and results obtained, is required. Minor comments:
- Introduction: Including a brief introduction to the spent culture medium methodology would be useful for the reader.
- Line 84: please define the abbreviation "MZT" as it is the first time used. - Line 97: please define the abbreviation "a.f." as it is the first time used. - Lines 101-106: rephrase this sentence to avoid using "with the exception of" due to the high number of parameters having significant differences (a quarter of the parameters included in Table 1). - Line 104: please define the abbreviation "HMG" as it is the first time used. - Figure 1: what do the authors mean with "OCC" and "MII"? "Number of..."? Please, clarify or refer to Table 1 in the legend. - Table 2: It is not clear for the reader which was the grouping criteria for the samples. Please, explain this issue in the legend of the Table in addition to Materials and Methods. - Lines 260-278: all this information is difficult to understand in text format, please consider arranging it into a Table. - Throughout the manuscript: The term "expression level of the sncRNA" in the spent culture medium seems to be inaccurate because it is not considering secretion/uptake rates or the contribution of the medium itself. Please, replace with "detected levels of the sncRNA" or any other appropriate term. -Discussion: additionally to the previous issue, a sentence discussing the correlation between extracellular and intracellular levels of sncRNAs should be added.Author Response
Dear Reviewer #1. I received your letter with comments of our manuscript “Small noncoding RNAs signature in determining the developmental potential of an embryo at the morula stage” (ijms-1011920). I appreciate the time you spent reviewing the manuscript and I am very grateful for your detailed analysis the data presented. Thank you very much for your opinion. I have corrected the article in the light of your comments and recommendations.
- English editing (Specialist edit) has been done (English Editing Invoice ID: english-24488);
- I have added two paragraphs to the Introduction section (lines 89-125 in the new version of the manuscript; highlighted in yellow), dedicated to the possible reasons for the lack of reproducibility of the obtained data when analyzing small noncoding RNAs in the culture medium of an embryo in the early stages of its development, citing published studies on this issue. In addition, I elaborated on what was done in our previous research in the field of embryology and what was the purpose of the present research;
- “Line 84: please define the abbreviation "MZT" as it is the first time used”
I have defined the abbreviation "MZT" as “maternal-to-zygotic transition” in the line 88 (new manuscript version);
- “Line 97: please define the abbreviation "a.f." as it is the first time used”
I have defined the abbreviation "a.f." as “after fertilization” in the line 134 (new manuscript version);
- “Lines 101-106: rephrase this sentence to avoid using "with the exception of" due to the high number of parameters having significant differences (a quarter of the parameters included in Table 1)”
I have rephrased this sentence as follows: “Comparison of clinical data and characteristics of the ART program revealed statistically significant differences in couples with pregnancy onset and live birth (n = 11) relative to couples with a negative result of the ART program (n = 16) in the following parameters: a higher AMH level (p = 0.0061), a shorter period of ovarian stimulation (p = 0.0226), lower doses of gonadotropin (p = 0.0101), and the preferable use of r-FSH (p = 0.0065) instead of human menopause gonadotrophins (HMG, p = 0.0025) for ovarian stimulation (Table 1)” (lines 137-142 in the new manuscript version).
- “Line 104: please define the abbreviation "HMG" as it is the first time used”
I have defined the abbreviation "HMG" as “human menopause gonadotrophin” in the line 142 (new manuscript version);
- “Figure 1: what do the authors mean with "OCC" and "MII"? "Number of..."? Please, clarify or refer to Table 1 in the legend”
I have clarified all parameters used in correlation analysis in Figure 1 legend in the lines 167-185.
- “Table 2: It is not clear for the reader which was the grouping criteria for the samples. Please, explain this issue in the legend of the Table in addition to Materials and Methods”
I have explained details of samples grouping in the legend of the Table 2 in the lines 220-223.
- Lines 260-278: all this information is difficult to understand in text format, please consider arranging it into a Table.
All information in lines 260-278 (old version of the manuscript) has been rewritten and arranged into Table 5 (lines 315-336 in new manuscript version).
- Throughout the manuscript: The term "expression level of the sncRNA" in the spent culture medium seems to be inaccurate because it is not considering secretion/uptake rates or the contribution of the medium itself. Please, replace with "detected levels of the sncRNA" or any other appropriate term.
I have done it where appropriate.
- - Discussion: additionally to the previous issue, a sentence discussing the correlation between extracellular and intracellular levels of sncRNAs should be added.
I have supplemented the "discussion" section with reasoning about extracellular and intracellular levels of sncRNAs.

Reviewer 2 Report
- There is considerable heterogeneity of the population participated in the study: different schemes for ovarian stimulation, one woman was oocyte donor, 4 patients with diminished ovarian reserves, differences in the sperm profile of male patients.
- Although the authors recognise that "IVF outcome depends on many factors such as patient age, genetics, reproductive system diseases, variability in the quality of oocytes and endometrium receptivity under the influence of exogenous gonadotropins, and sperm fertility" (lines 292-294) the effect of these factors on the levels of sncRNAs from embryonic culture media was not evaluated.
- The results were analysed according only to the morphological quality of the blastocysts and not to the pregnancy and live birth rates as the authors state in the first sentence of the abstract "the aim of the study was the identification of the key small noncoding RNA (sncRNA) molecules that participate in maternal-to-zygotic transition and determine development potential and competence to form a healthy fetus" (lines 25-28).
- It is not clear if the embryos were cultured in groups or in single.
- Articles 12 & 13 are relatively old; there are more recent references.
- It is not clear if the study was approved by an Institutional Ethical Committee and if the patients gave an informed consent
Author Response
Dear Reviewer #2. I received your letter with comments of our manuscript “Small noncoding RNAs signature in determining the developmental potential of an embryo at the morula stage” (ijms-1011920). I appreciate the time you spent reviewing the manuscript. I have corrected the article in the light of your comments and recommendations.
- English editing (Specialist edit) has been done (English Editing Invoice ID: english-24488);
- There is considerable heterogeneity of the population participated in the study: different schemes for ovarian stimulation, one woman was oocyte donor, 4 patients with diminished ovarian reserves, differences in the sperm profile of male patients. Although the authors recognise that "IVF outcome depends on many factors such as patient age, genetics, reproductive system diseases, variability in the quality of oocytes and endometrium receptivity under the influence of exogenous gonadotropins, and sperm fertility" (lines 292-294) the effect of these factors on the levels of sncRNAs from embryonic culture media was not evaluated. The results were analysed according only to the morphological quality of the blastocysts and not to the pregnancy and live birth rates as the authors state in the first sentence of the abstract "the aim of the study was the identification of the key small noncoding RNA (sncRNA) molecules that participate in maternal-to-zygotic transition and determine development potential and competence to form a healthy fetus" (lines 25-28).
In the present study, we did not aim to analyze the influence of the parameters of the ovarian stimulation protocol and the clinical data of patients on the implantation potential of the embryo. We focused on the differences in the molecular profile of morula, which has different potential for further development, regardless of the parameters mentioned above. But starting from the embryological stage, all conditions for performing the research were identical. It was important for us to find among the target genes of small noncoding RNAs that determine the high potential for morula development, those that are involved in MZT and blastulation. To do this, we used various algorithms to search for sncRNA potential target genes followed by their subsequent identification among those genes whose function has already been experimentally proven by other researchers.
- It is not clear if the embryos were cultured in groups or in single.
Each embryo was cultured in single (Figure 4)
- Articles 12 & 13 are relatively old; there are more recent references.
I added references 14-17
- It is not clear if the study was approved by an Institutional Ethical Committee and if the patients gave an informed consent
This information is presented in lines 441-444 of the new version of the manuscript.

Reviewer 3 Report
The current research paper has some interesting results in identifying potential markers for embryo quality. However, some minor and major questions to be answered before accepting this manuscript. Besides I see a lot of similarities between the current version of the paper and the previously published paper entitled “Cell-Free, Embryo-Specific sncRNA as a Molecular Biological Bridge between Patient Fertility and IVF Efficiency” which makes this manuscript less interesting for the readers
The current version needs a lot of corrections in the experimental design, authors need to present a proper experimental design and control data.
Minor and major
Line 33; provide a full form of piRNAs
Line 43: authors can’t NGS as a keyword, provide either full form or remove NGS from the keyword
Line 149: Check the spacing in the title
Line 292: Never start a sentence with an Abbreviation change IVF to in vitro fertilization, Please check in the whole manuscript.
Line 397-432: Authors need to provide proper experimental design in a schematic way so that readers can understand better, in the current form it is not clear how the experiments were performed
There is no clear information on how many technical/biological replicates were used to perform total RNA extraction followed by miRNA sequencing
Line 422-423: Did the spent medium is pooled or collected separately based on the embryo stage?
Line 433-437: It is quite surprising that authors could extract total RNA from 25 uL of embryo culture medium, which is not believable and makes all the data questionable? Did the authors perform bioanalyzer to check the total RNA concentration and quality of RNA? if so authors need to present this data.
Did authors perform total RNA extraction from blank conditioned, if so what are the results of miRNA sequencing?
Author Response
Dear Reviewer #3. I received your letter with comments of our manuscript “Small noncoding RNAs signature in determining the developmental potential of an embryo at the morula stage” (ijms-1011920). I appreciate the time you spent reviewing the manuscript. Thank you very much for your opinion. I have corrected the article in the light of your comments and recommendations.
- “Besides I see a lot of similarities between the current version of the paper and the previously published paper entitled “Cell-Free, Embryo-Specific sncRNA as a Molecular Biological Bridge between Patient Fertility and IVF Efficiency” which makes this manuscript less interesting for the readers”
This study is fundamentally different from the previous one. This is described in the “Introduction” section in lines 109-124 (new version of the manuscript).
In our previous study [23], we sequenced sncRNAs in the spent culture medium of an excellent blastocyst and in its blastocoelic fluid and showed the prevalence of piRNA molecules over miRNAs in the analyzed sample contents. Moreover, we have demonstrated significant differences in the detected levels of the sncRNA in spent culture media from embryos on the 4th day after fertilization, which had different rates of development, and by the fifth day, reached either the 3–7-cell embryo stage, or were morula, cavernous morula, or excellent, good, fair, and poor-quality blastocysts. In the present study, we were interested in comparing the levels of miRNAs and piRNAs in the spent culture media from embryos that had the same rates of development and reached the morula stage by the fourth day but had different developmental outcomes by the fifth day (degeneration or developmental arrest, a blastocyst of poor quality, or a blastocyst of excellent quality which, when implanted in the uterine, led to the birth of a healthy child). The choice of the morula stage for a detailed study was determined by the possibility of assessing, on the one hand, the ability of the embryo to pass the 8-cell stage at which the main MZT wave occurs and, on the other hand, the potential of the embryo to develop into the blastocyst able to be implanted for initiating pregnancy. We found it interesting to compare the list of potential target genes of the miRNAs and piRNAs that we identified with the data published by other colleagues on the expression profile of these target genes, the protein products of which are involved in MZT.
- English editing (Specialist edit) has been done (English Editing Invoice ID: english-24488);
- “The current version needs a lot of corrections in the experimental design, authors need to present a proper experimental design and control data”.
Flow diagram of the experimental design is presented in “Materials and methods” section, “Experimental Design” subsection, Figure 4.
- Line 33; provide a full form of piRNAs
Full forms of piRNAs were given according to the names in piRNABank (http://pirnabank.ibab.ac.in/) and and I haven't made any changes
- Line 43: authors can’t NGS as a keyword, provide either full form or remove NGS from the keyword
I have substituted NGS for small RNA deep sequencing in line 43 (new manuscript version)
- Line 149: Check the spacing in the title
It has been done.
- Line 292: Never start a sentence with an Abbreviation change IVF to in vitro fertilization, Please check in the whole manuscript.
It has been done.
- Line 397-432: Authors need to provide proper experimental design in a schematic way so that readers can understand better, in the current form it is not clear how the experiments were performed
Flow diagram of the experimental design is presented in “Materials and methods” section, “Experimental Design” subsection, Figure 4.
- There is no clear information on how many technical/biological replicates were used to perform total RNA extraction followed by miRNA sequencing
Isolation of RNA was performed once from a 25 µl column eluate using the miRNeasy Serum / Plasma Kit, Qiagen. One third of the column eluate was used for small RNA sequencing, another third of the column eluate was used for RT-PCR.
- Line 422-423: Did the spent medium is pooled or collected separately based on the embryo stage?
The spent culture medium wasn’t pooled but collected separately based on the embryo stage
(Flow diagram of the experimental design is presented in “Materials and methods” section, “Experimental Design” subsection, Figure 4)
- Line 433-437: It is quite surprising that authors could extract total RNA from 25 uL of embryo culture medium, which is not believable and makes all the data questionable? Did the authors perform bioanalyzer to check the total RNA concentration and quality of RNA? if so authors need to present this data.
We extracted total RNA not only from 25 µl of embryo culture medium but from several nanoliter of blastocoelic fluid (the results of small RNA sequencing were presented in our previous study [23]). It is impossible to reliably measure the total concentration of RNA in spent culture medium and assess its quality by Nanodrop spectrophotometer, Qubit fluorometer, or Agilent Bioanalyzer system which also applies to other biological fluids such as, for example, human peripheral blood plasma [26], which has prompted the search for normalizing endogenous RNA suitable for accurately quantitating microRNAs.
In the present study, the relative expression of sncRNA in the embryo culture medium was determined using the ∆∆Ct method using hsa_piR_023338 (DQ601914, GenBank, available online: https://www.ncbi.nlm.nih.gov/genbank/) as the reference RNA and culture medium without contact with any embryo incubated for 4 days at 37 °C as a reference sample to calculate the fold change of expression level in a sample. hsa_piR_023338 was chosen as the reference RNA due to its consistent expression level in all 51 analyzed samples including reference sample.(described in Materials and Methods).
- Did authors perform total RNA extraction from blank conditioned, if so what are the results of miRNA sequencing?
Yes, we performed total RNA extraction from blank conditioned medium. The results of sncRNA sequencing are presented in Table 2 (Reference #207).

Round 2
Reviewer 2 Report
Nothing to mention
Reviewer 3 Report
The current version of the manuscript has been improved a lot and it can be accepted in the present form.